# Dimerization of Acetylene to Monovinylacetylene (MVA) by Bimetallic Zr/Cu Catalyst in Nieuwland Catalytic System

**DOI:** 10.3390/molecules27030602

**Published:** 2022-01-18

**Authors:** Leng Zheng, Ruolin Lin, Dingjie Luo, Liang Guo, Jinli Zhang

**Affiliations:** 1School of Chemistry and Chemical Engineering, Key Laboratory for Green Processing of Chemical Engineering of Xinjiang Bingtuan, Shihezi University, Shihezi 832003, China; zfg1523@163.com (L.Z.); linruoling163@163.com (R.L.); dingjieluo@163.com (D.L.); 2School of Chemical Engineering and Technology, Tianjin University, Tianjin 300072, China

**Keywords:** Nieuwland catalyst, acetylene dimerization, monovinylacetylene, Cu-based catalyst, bimetallic catalyst

## Abstract

Nieuwland catalyst is a key step in the dimerization of acetylene. Various zirconium metal additives incorporating Nieuwland catalysts were prepared, and their catalytic performances were assessed in acetylene dimerization. Different characterization techniques (i.e., thermogravimetric analysis, temperature-programmed reduction, X-ray diffraction, X-ray photoelectron spectroscopy, hydrogen ion concentration measurement and transmission electron microscopy) were employed in this study. The best catalytic performance was obtained over zirconium-acetylacetonate-incorporated Nieuwland catalysts, with an acetylene conversion of 53.3% and a monovinylacetylene selectivity of 87.4%. Based on these results, the zirconium acetylacetonate additive could reduce the types of transition state complexes, and it could also change the morphology of the catalyst. In addition, the additives could significantly inhibit the occurrence of trimerization products and polymers. Hence, the conversion of acetylene, monovinylacetylene selectivity, and stability of the Nieuwland catalysts were enhanced.

## 1. Introduction

Chloroprene rubber (CR), which possesses excellent physical and chemical properties, is one of the five engineering plastics and is widely used in various applications, such as industrial applications, agricultural activities, and daily life [1,2,3,4]. Generally, two methods are currently employed for the industrial production of CR. One is a butadiene-based method, and the other is an acetylene-based method [5,6,7]. With the gradual development of coal resources, the acetylene-based method has received extensive attention in China because of the coal resource abundance [8]. Dimerization of acetylene to monovinylacetylene is the key step in the acetylene-based method, and the Nieuwland catalyst (NC) is often used with composites of CuCl-NH_4_Cl (or KCl)-HCl-H_2_O. However, it is necessary to address the following challenges. The conversion rate of acetylene and the selectivity of MVA are low, and the contact between acetylene and an active component within a gas–liquid reactor is uneven. The solubility of acetylene in water is poor, and this can lead to an inadequate gas content in the liquid. Furthermore, the duration of the contact between the reactants and active components is short. The poor contact between acetylene and the catalyst is further aggravated by the presence of by-products, which can easily cover the surfaces of the active components of the catalyst. To mitigate this issue, Cu^+^ can catalyze the dimerization reaction, hydrochlorination, and acetaldehyde reaction, which allows the continual dimerization of acetylene to MVA. This reaction can produce divinyl acetylene (DVA) and other polymers. Thus, the focus of this research is to find a catalyst that can achieve (1) good contact between acetylene and the active components, (2) reduce the occurrence of side reactions, and (3) increase the effective utilization rate of acetylene during the dimerization reaction.

In recent years, the reaction system has been continuously optimized by numerous researchers. Various strategies, such as adding a second metal to develop a synergistic catalyst, ligand modification, catalytic mechanism, and process strengthening, have been proposed, with encouraging results reported. Use isotope labeling and other methods to study the reaction mechanism and explore the cause of catalyst deactivation [9,10]. From the perspective of process enhancements, electric field enhancement [11], high-shear strengthening [12], and the incorporation of graphene can improve the conversation of C_2_H_2_ in acetylene dimerization [13]. It has been reported that MVA selectivity can be enhanced by using a bimetallic synergistic catalyst (adding a second metal) [14,15,16]. Ligand modification has been reported to be effective for improving the catalytic activity and stability of the catalyst after adding ligands [17,18,19,20]. Changing the catalytic system (i.e., replacing the aqueous Nieuwland catalyst with an organic phase) can significantly increase the conversion rate of acetylene [16,21,22]. The catalyst system can also be changed from a gas–liquid bubbling fluidized bed reactor to a gas–solid phase reactor [23], which can address the issue of catalyst recovery but deactivated quickly. In addition, side-reactions of acetylene hydration and acetylene dimerization often occur in the Nieuwland catalyst, which results in a low selectivity of MVA.

Although these studies found that the addition of a second metal can significantly improve the selectivity of MVA (90%) and the stability of the catalyst in the acetylene dimerization reaction, the conversion rate of acetylene was reduced significantly (10–15%). Therefore, we seek a metal promoter that can change the morphology of the catalyst and enhance the conversion rate of acetylene. In this work, the addition of zirconium metal to the catalyst was found to promote the MVA selectivity and conversion of C_2_H_2_ through a series of comparisons, because a sufficient bubbling effect strengthened the contact between acetylene and the catalyst.

## 2. Experimental

### 2.1. Materials

C_2_H_2_ (gas, 99.5%) and N_2_ (gas, 99.998%) were purchased from Shanghai Weichuang Standard Gas Analytical Technology Co., Ltd (Shanghai, China). Zirconium nitrate pentahydrate (Zr_1_), zirconium butoxide (Zr_2_), zirconium oxychloride octahydrate (Zr_3_), zirconium chloride (Zr_4_), zirconium acetylacetonate (Zr_5_), acetylacetone (C_1_), acetone (C_2_), acetic acid (C_3_), NH_4_Cl, and CuCl, were purchased from Macklin Chemical Reagent Co., Ltd (Xi’an, China) and all reagents were of analytical grade directly used without purification. Deionized water was prepared in the laboratory using a standard reagent-type ultrapure water machine (FBZ1002-SUP, Qingdao Flom Technology Co., Ltd., Qingdao, China).

### 2.2. Catalyst Preparation

The Nieuwland catalyst used as a control was labeled NC, and the catalysts containing Zr_X_ additive were denoted as Zr_X_/NC, where X denotes the different kinds of anions added. The amount Zr_X_ added was calculated in moles based on the amount of CuCl added.

The Nieuwland catalysts were prepared by dissolving 5.35 g (0.1 mol) of NH_4_Cl and 9.9 g (0.1 mol) of CuCl in 10 mL of deionized water at 80 °C, and a reddish-brown homogeneous mixture was obtained. A certain molar amount of Zr_X_ was then added to the NC, which varied based on the different metal additives, to achieve a desired molar quantity of zirconium ions. Before the reaction, the reactor was purged with nitrogen to eliminate the air. Stirring continued for at least 15 min until the solids had completely dissolved and the expected catalyst was obtained. The fresh and used NC and Zr_5_/NC were left in the refrigerator at 3 °C for 6 h (the NC and Zr_5_/NC after 8 h of reaction are referred to as used NC and Zr_5_/NC, respectively). They were then filtered from the catalytic solution, washed with water, and dried in a vacuum oven at 80 °C to obtain a solid catalyst that was ready for characterization.

### 2.3. Catalytic Activity Analysis and Evaluation

The catalysts were tested in a self-designed bubble column reactor (length: 400 mm, outer diameter: 40 mm, inner diameter: 10 mm) that was made of glass, which was used for the acetylene dimerization reaction. Pure acetylene was then passed through the reactor. After the reaction product was dried, the gas phase products were analyzed on-line using a GC-2014C gas chromatograph (Shimadzu Corporation, Tokyo, Japan) immediately.

The conversion of acetylene (X_A_) and the selectivity of the products (S_MVA_) were calculated as follows:$$X_{A}=\frac{\varphi_{1}+2\varphi_{2}+2\varphi_{3}+3\varphi_{4}}{\varphi_{1}+2\varphi_{2}+2\varphi_{3}+3\varphi_{4}+\varphi_{5}}\times 100\%,\;S_{\mathrm{MVA}}=\frac{2\varphi_{2}}{\varphi_{1}+2\varphi_{2}+2\varphi_{3}+3\varphi_{4}}\times 100\%,$$
where *φ_1_, φ_2_, φ_3_, φ_4_* and *φ_5_* are the volume fractions of the outlet composition, CH_3_CHO, MVA, Chloroprene (CP), and Divinylacetylene (DVA), C_2_H_2_, respectively.

### 2.4. Catalyst Characterization

The pH of the catalyst was measured using a pen-like PHB-3 meter (Shanghai Sanxin Instrument Factory, Shanghai, China). 

The metal contents in the catalysts were determined using inductively coupled plasma–atomic emission spectroscopy (ICP-OES, Agilent 5110, Agilent Technologies, Inc., Santa Clara, CA, USA). 

Thermogravimetric analysis (TGA) and differential scanning calorimetry (DSC) of the samples were performed using a NETZSCH STA 449F3 Jupiter thermogravimetric–differential scanning calorimeter and a simultaneous thermal analyzer (NETZSCH Group, Selb, Germany) in a nitrogen atmosphere at a flow rate of 30 mL min^−1^. The temperature was increased from 50 to 900 °C at a heating rate of 10 °C min^−1^.

Temperature-programmed reduction (TPR) experiments (ASAP 2720, Mike Instruments ltd., Georgia, US) were performed in a micro-flow reactor fed with a 10.0% H_2_-Ar mixture flowing at a rate of 120 mL min^−1^. Prior to each test, the samples were treated with N_2_ gas at 80 °C for 30 min. The temperature was then increased from 80 to 900 °C at a heating rate of 10 °C min^−1^, and the temperature was held at 900 °C for 0.5 h. 

X-ray diffraction (XRD) measurements were performed using a Bruker D8 ADVANCE X-ray diffractometer (Bruker Company, Bremen, Germany) equipped with a Cu Kα X-ray tube operating at 40 kV and 40 mA in the 2θ scan range between 10° and 90°.

X-ray photoelectron spectroscopy (XPS) spectra were recorded using a Kratos Axis Ultra DLD spectrometer (Kratos Company, Manchester, UK) employing a monochromated Al-Kα X-ray source, hybridoptics, and a multi-channel plate and delay line detector. The peak area corresponded to its valence content.

The liquid catalysts were analyzed by transmission electron microscopy (TEM). A drop of the liquid catalyst was allowed to evaporate on a holey carbon film supported by a 300-mesh copper TEM grid. Bright-field and annular darkfield imaging experiments were carried out using a JEM2100F transmission electron microscope (Nippon Electronics Co., Ltd., Tokyo, Japan).

## 3. Results and Discussion

### 3.1. Results

Previously, we have reported that the addition second metal can improve the performance of NC in acetylene dimerization [14,15,16]. Inspired by these results, we test the catalytic activities through six types of catalysts presented in Figure 1. Although good acetylene dimerization activity was observed with the use of NC, the selectivity for MVA was relatively poor. Interestingly, after incorporating a second metal component, the catalytic selectivity for MVA was significantly enhanced. The selectivity to MVA of catalysis were in the order of Zr_5_/NC > Zr_1_/NC > Zr_4_/NC > Zr_2_/NC > Zr_3_/NC > NC.

It is worth noting that NC, without the addition of zirconium acetylacetonate, showed a relatively stable activity. However, its selectivity for MVA exhibited a downward trend, and it remained at a low level of about 68% (Figure 1b). The ZrX/NC catalyst exhibited a relatively better selectivity for MVA with the addition of 0.001 mol zirconium metal additives. Among them, the zirconium-acetylacetonate-incorporated NC (denoted as Zr_5_/NC) catalyst exhibited the best performance, with an almost 87.4% MVA selectivity and a C_2_H_2_ conversion greater than 53.3% after 8 h. The zirconium-nitrate-pentahydrate-incorporated NC (denoted as Zr_1_/NC) showed a similar catalytic performance to that of Zr_5_/NC. 

This result may have been due to the Zr^4+^ ions, as the NO_3_^−^ and Cu^+^ did not coexist under acidic conditions, and acetylacetone thermally decomposed at the reaction temperature. However, for the zirconium-oxychloride-octahydrate-incorporated NC (denoted as Zr_3_/NC) and the zirconium-chloride-incorporated NC (denoted as Zr_4_/NC) under similar conditions, a more restrained effect on the conversion of C_2_H_2_ was observed. Such a result may have been due to the greater amount of Cl^-^ brought in Nieuwland catalyst system, which lowered the C_2_H_2_ conversion. 

Since Zr_5_ does not possess stable chemical bonds, it can easily be decomposed to acetylacetone (C_1_), acetone (C_2_), and acetic acid (C_3_) in the catalytic system. Hence, to verify the effects of the Zr_5_ promoter ligand on the reaction, C_1_, C_2_, and C_3_ were analyzed. The catalytic performance of C_X_/NC under similar reaction conditions to those used for acetylene dimerization is shown in Figure 2. C_2_/NC and C_3_/NC showed no significant performance enhancements. C_1_/NC showed a good activity of 37.0% and a relatively good MVA selectivity of 80.3% after 480 min. During the reaction, it is hypothesized that coordination between C_1_ and the active center of the catalyst may have occurred. As such, this may have reduced the amount of DVA produced due to MVA trimerization, therefore improving the catalytic activity and selectivity of the catalyst.

### 3.2. Characterization and Analysis of Catalyst

To determine the nature of the Zr_5_/NC catalyst, detailed structural and chemical analyses were conducted using a combination of characterization techniques.

#### 3.2.1. Composition of Catalyst

The TGA and (differential thermogravimetry) DTG curves of the fresh and used NC and those of the Zr_5_/NC catalysts are shown in Figure 3a,b, respectively. According to Figure 3a, neither the fresh nor the used NC catalyst showed observable mass losses before reaching 150 °C. This result indicated that there was a small amount of water adsorbed on the surface of the catalyst. Within the temperature range of 150–400 °C, there was a gradual mass loss for the fresh NC catalyst (whereby it reached 30.87%), and this was attributed to the loss of NH_4_Cl. As the temperature exceeded 400 °C, there was a rapid decrease in the mass of the catalyst due to the decomposition of CuCl. In contrast, the used NC catalyst underwent a significant weight loss of 26.25% in the temperature range of 150–400 °C, which was attributed to the loss of NH_4_Cl and C_2_H_2_. At the temperature range of 150–400 °C (with the majority of the mass loss in the range of 150–320 °C), the mass losses observed for both the fresh and used NC catalysts were mainly due to the decomposition of NH_4_Cl.Upon the incorporation of Zr_5_, there was a shaper drop in the mass for both the fresh and used Zr_5_/NC (Figure 3b) in the temperature range of 150–340 °C, which was attributed to the coordination of C_3_ to the active center. When the temperature exceeded 475 °C, there was a significant mass loss in the catalyst, which was mainly due to the decomposition of CuCl. The comparison of the results in Figure 3a,b shows that the decomposition temperature of NH4Cl was lower after the addition of Zr5, while the decomposition temperature of CuCl increased. According to the TGA and DTG curves, Zr_5_ may have promoted the stability of the NH_4_Cl and CuCl, which accounted for the reduced mass loss in the range of 150–400 °C and the higher decomposition temperature of CuCl.

#### 3.2.2. Effects of Additional Components on Catalytic Performance of NC Catalyst

The H_2_-TPR profile of the fresh NC catalyst revealed two characteristic reduction bands between 400 °C and 696 °C, the Cu^2+^ and Cu^+^ in the Cl^−^ rich environment are harder to reduce, as copper ions can form complexes with chloride ions to improve anti-reduction stability, respectively (shown in Figure 4a) [23,24,25,26,27]. After the reaction (used NC catalyst), the C_2_H_2_ destroy the complex formed by copper ion and chloride ion, reduce its reduction temperature, lead to the peak of Cu^2+^ and Cu^+^ shifted left. The H_2_-TPR profile of used NC, Zr_5_/NC and used Zr_5_/NC shows another reduction peak at 400–550 °C caused by free copper ions. 

The results in Figure 4a,b show that the widths of both the Cu^+^ and Cu^2+^ reduction peaks became narrower after the addition of Zr_5_, it is reasonable to conclude that the complexes in the Zr_5_/NC catalyst system were purer, which was consistent with the conclusions drawn from the TGA results. The used catalyst may have formed a specific complex with acetylene, which reduced its reduction temperature. 

Previously report shown pH of the catalyst solution have an effect on catalytic activity [28]. As shown in Table 1, Zr_5_/NC possessed a lower initial pH value than NC, and the pH of both catalytic systems remained relatively stable with time. This result was due to the instability of Zr_5_ in the Nieuwland catalytic system, and the catalyst could be easily decomposed to produce acidic C_3_, which could reduce the initial pH of the Zr_5_/NC solution.

#### 3.2.3. Change in Valence of Active Component during Reaction

Figure 5 shows the XRD patterns of the fresh and used NC and Zr_5_/NC catalysts. The result shown that the main component between them is Cu_112_Cl_168_, indicating that Cu_112_Cl_168_ may be the active component in the catalyst. With the addition of Zr_5_, other impurity peaks in the catalyst decrease, such as 2θ = 30.38, and the XRD pattern of Zr_5_/NC was more consistent with Cu_112_Cl_168_.

X-ray photoelectron spectroscopy can provide the chemical state and distribution of the active component (i.e., Cu) in the catalyst before and after the reaction were shown in the Figure 6a,b [29,30]. Thus, Figure 6a curves were deconvoluted to determine the ratio of each Cu species in the catalyst. During the preparation and storage of the catalysts, Cu^+^ could be reduced to Cu^2+^. As such, the active component Cu in the catalyst was composed of a large amount of Cu^+^ and a small amount of Cu^2+^ (CuCl and HCl will react in hot water: CuCl + HCl = H[CuCl₂]). For the fresh NC and Zr_5_/NC, the contents of Cu^+^ were 76.27 and 71.20%, respectively (as shown in Table 2), while the contents of Cu^+^ in the used NC and Zr_5_/NC catalysts were 72.78% and 68.64%, respectively. Combining the results of XRD, it is found that the closer the content of Cu^+^/Cu^2+^ is to Cu_112_Cl_168_, the better the catalytic activity. The Figure 6b shown the existence of different elements in the catalysis. In addition to the surface characterization by XPS analysis, the result obtained from ICP-OES also indicated that the loss ratio of Cu after the reaction was small. 

#### 3.2.4. Morphology and Structure of Catalysis

Figure 7 shows the morphological characteristics of the catalyst after starting the reaction and 8 h after the reaction. Numerous black polymers can be clearly observed in the NC catalyst in the reaction tube after eight hours of reaction. In contrast, no precipitate was evident in the reaction tube with Zr_5_/NC. This observation verified that the addition of Zr_5_ could reduce the production of polymer and the number of active centers. It is worth noting that a perfect bubbling effect could not be achieved due to the highly viscous nature of the catalyst, which resulted in uneven contact between the active components and the reactants. As shown in Figure 7c,d, after adding Zr_5_, the system was more likely to form bubbles in the solution under the same reaction conditions. As such, this reduced the residence time of acetylene in the catalyst, increased the contact area between these two components, reduced the formation of polymers, and improved the catalytic performance of the catalyst. 

Transmission electron microscopy (TEM) was conducted to investigate the morphology and structure of the catalyst after starting the reaction (Figure 8a,c) and after reacting for 8 h (Figure 8b,d). During the initial stage of the reaction, the active components were uniformly dispersed in the solvent for both NC and Zr_5_/NC, and no black polymer was observed (Figure 8a,c). However, after reacting for 8 h, morphological changes were evident for NC, whereby severe agglomeration was observed (Figure 8b), as MVA stuck to the active centers and continued to polymerize with acetylene. However, there was no evident agglomeration for the Zr_5_/NC catalyst after reacting for 8 h, which may have been attributed to the addition of Zr_5_. The resultant high polymer was attached to the surfaces of the active components, which could affect the contact between acetylene and the catalyst. This could then reduce the performance of the catalyst. The conclusion obtained from the TEM results was consistent with that obtained from the above-mentioned analyses.

## 4. Conclusions

The catalysis of acetylene dimerization over Zr_5_/NC catalysts showed that the addition of a second metal Zr component in the catalyst could significantly enhance its MVA selectivity. The best catalytic performance was obtained over the Zr_5_/NC catalyst, with an acetylene conversion of 53.3% and MVA selectivity of 87.4%. The addition of Zr_5_ can promote the formation of Cu_112_Cl_168_ and make the active components of the catalyst purer, thereby promoting the improvement of the catalytic performance of the catalyst.

## Figures and Tables

**Figure 1 molecules-27-00602-f001:**
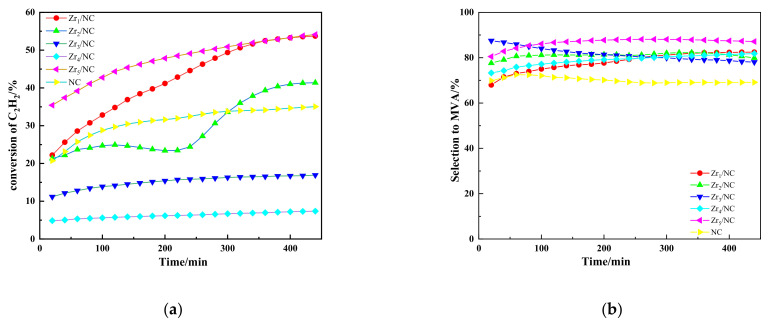
Catalytic performance of NC and Zr_X_/NC. (**a**) C_2_H_2_ conversion with time on stream, and (**b**) MVA selectivity with time on stream.Reaction conditions; temperature (T) = 80 °C and C_2_H_2_ gas hourly space velocity (GHSV) = 110 h^−1^.

**Figure 2 molecules-27-00602-f002:**
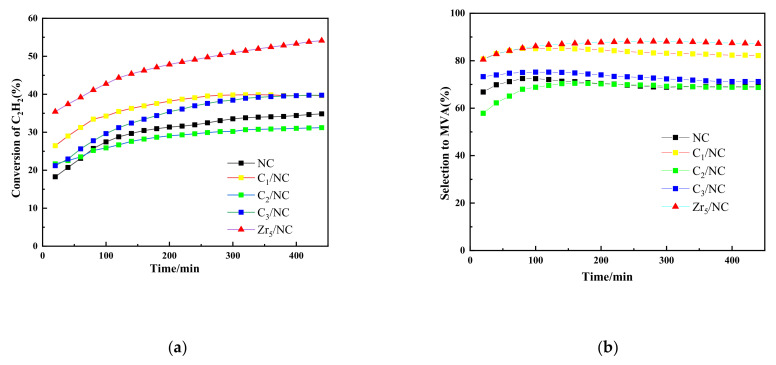
Catalytic performance of NC and C_X_/NC. (**a**) C_2_H_2_ conversion with time on stream, and (**b**) MVA selectivity with time on stream. Reaction conditions; temperature (T) = 80 °C, C_2_H_2_ gas hourly space velocity (GHSV) = 110 h^−1^.

**Figure 3 molecules-27-00602-f003:**
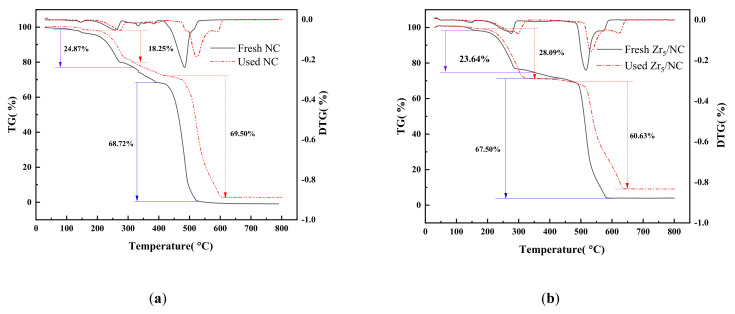
TGA and DTG curves of catalysts. (**a**) TG result of NC, and (**b**) TG result Zr_5_/NC.

**Figure 4 molecules-27-00602-f004:**
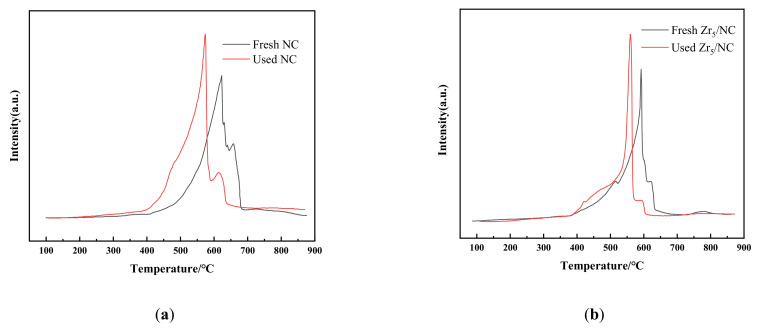
H_2_-TPR profiles. (**a**) The intensity signal with temperature of NC, and (**b**) The intensity signal with temperature of Zr_5_/NC.

**Figure 5 molecules-27-00602-f005:**
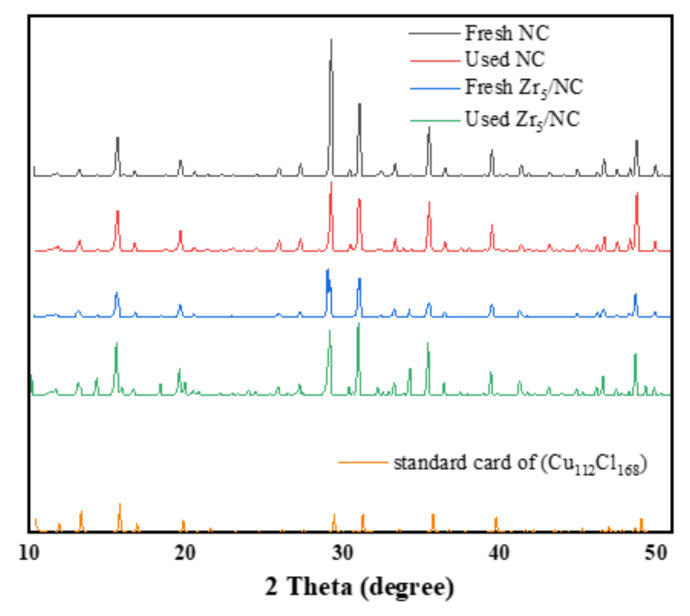
XRD spectra. Standard card of Cu_112_Cl_168_: Cambridge Crystallographic Data Centre (CCDC) and the deposition number is 2092361.

**Figure 6 molecules-27-00602-f006:**
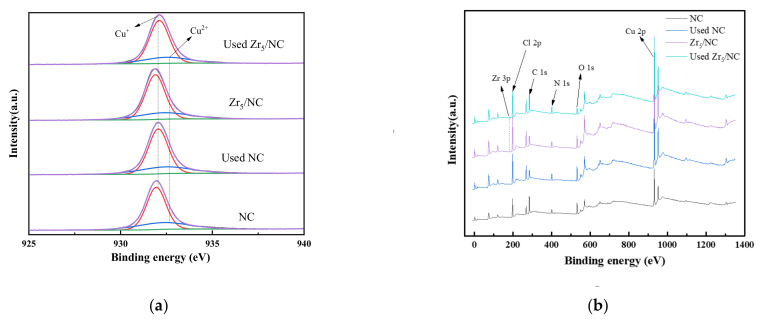
XPS spectra of catalysts, (**a**) the Cu2p spectra and (**b**) XPS survery spectra.

**Figure 7 molecules-27-00602-f007:**
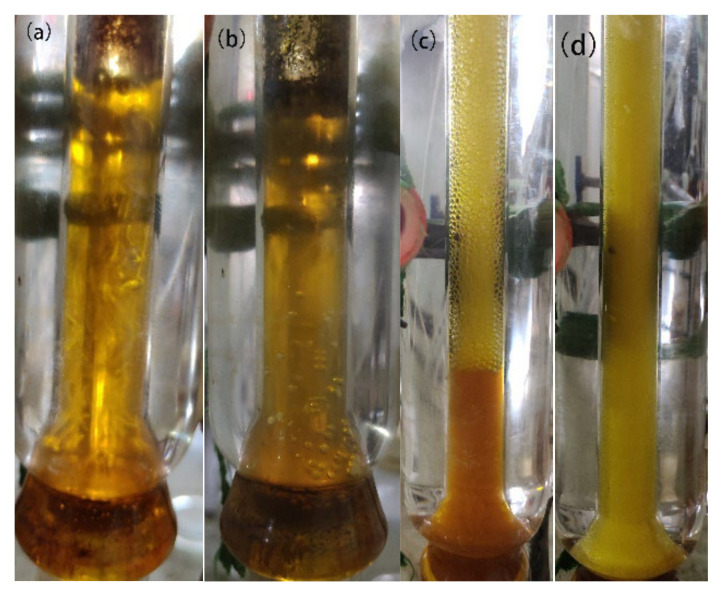
Photographic images of the bubble column reactors containing NC (**a**,**b**) and Zr_5_/NC (**c**,**d**) before and after the reaction.

**Figure 8 molecules-27-00602-f008:**
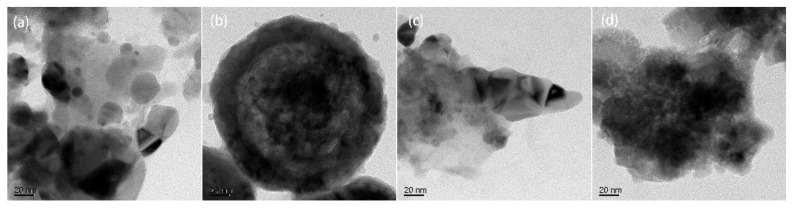
TEM images. (**a**) the fresh NC, (**b**) the used NC, (**c**) the fresh Zr_5_/NC, and (**d**) the used Zr_5_/NC.

**Table 1 molecules-27-00602-t001:** pH of NC and Zr_5_/NC at different time intervals during the acetylene dimerization reaction.

Sample	pH							
	1 h	2 h	3 h	4 h	5 h	6 h	7 h	8 h
NC	1.2	1.2	1.3	1.2	1.2	1.2	1.3	1.3
Zr_5_/NC	1.1	1.1	1.2	1.2	1.2	1.3	1.2	1.3

**Table 2 molecules-27-00602-t002:** Relative contents and binding energies of Cu^+^ and Cu^2+^ in the fresh and used catalysts as determined by XPS and ICP-OES (^a^ denotes the determination using XPS, and ^b^ denotes the determination using ICP-OES).

Sample	Area% (Cu) ^a^		Cu% (Metal Ion Content) ^b^	Cu Loss (%) ^b^
	Cu^+^	Cu^2+^		
Fresh NC	76.27	26.73	40.77	
Used NC	72.78	27.22	40.35	1.03
Fresh Zr_5_/NC	71.20	28.80	38.46	
Used Zr_5_/NC	68.64	31.36	37.19	3.30

## Data Availability

Not applicable.

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
