# Peer review of "Dimerization of Acetylene to Monovinylacetylene (MVA) by Bimetallic Zr/Cu Catalyst in Nieuwland Catalytic System"

_molecules, 2022, doi:10.3390/molecules27030602_

Round 1
Reviewer 1 Report
Title «Dimerization of acetylene to monovinylacetylene (MVA) by bimetallic Zr/Cu catalyst in Nieuwland catalytic system»
This manuscript devoted to the investigation of various zirconium metal additives incorporating Nieuwland catalysts. The topic is quite interesting and the catalytic results shown the synergistic effect when the Zr added to the catalysts. Although the catalysts have been characterized by a fruitful combination of the physical methods the evidence of Zr additive behavior from my opinion was not so powerful and reasonable, so it isn't recommended this manuscript to be published in Molecules journal.
The comments for authors:
- From the Introduction part does not really clear why the Zr-containing additives were chosen.
- As I understand the catalysts under reaction conditions is dissolved into solution, then for the investigation the catalyst was filtered from the catalytic solution, washed with water, and dried in a vacuum oven at 80°C to obtain a solid catalyst that was further used for characterization. The main question is what the authors investigate using the different physical methods? How to prove that the data obtained for dried catalysts could be directly used in order to make the conclusions concerning the structure and chemical composition of it in dissolved state?
- Authors carried out temperature-programmed reduction with a 10.0% H2-Ar mixture flowing at a rate of 120 mL min−1. H2-TPR profile revealed two characteristic reduction bands between 400 and 696°C, with peaks centered at 622.4 and 655.2°C. They claim that those two peaks correspond to reduction of Cu(II) to Cu(I) and Cu(I) to Cu(0). However, there is some additional shoulder at around 450°C for the fresh NC sample. It is worth to explain which species cause such shoulder. On the other hand, authors do not refer to any other article where the same reduction temperatures for Cu species shown. In [ Kühl, A. Tarasov, S. Zander, I. Kasatkin, M. Behrens, Chem. Eur. J. 2014, 20, 3782 – 3792] it was shown that there are two peaks for the reduction of Cu-containing catalyst at 245 and 285°C performed in 5 vol.% H2 in Ar with a flow of 80 mL min−1. On the other hand, effect of different parameters on TPR profiles was investigated in [Altamira Notes Vol. 3.1]. It was shown that increase of H2 concentration as well as flow rate shift reduction temperature to lower values. Thus, authors should explain such high reduction temperature of Cu species in NC catalyst. Besides, Nieuwland catalysts were prepared by dissolving of NH4Cl and CuCl. How can authors explain the appearance of Cu(II) species in the catalysts?
- In Experimental part the Fourier-transform infrared (FT-IR) spectroscopy mentioned as one of the methods used for the catalysts characterizations. However in Results and Discussion part there is absence of the any data obtained by FT-IR.
- Concerning the XRD data - The XRD phase analysis was performed somewhat carelessly. Apart from very small details of the XRD patterns (no indices, small font size, a few reflections marked) there are some discrepancies between the author’s analysis and actual XRD data. For instance, the authors suggest the existence of "(NH4Cl)CuCl3 and (NH4Cl)2CuCl3"(looks like a typo from (NH4)CuCl3 PDF #88-2347 and (NH4)2CuCl3 PDF #74-412) in the fresh NC catalyst. Yet there is no the most intensive reflections of (NH4)CuCl3 PDF #88-2347 from lower angles 011 (2q=11.7°) and 020 (2q=12.5°) or any explanation of their absence. The same is for (NH4)2CuCl3 PDF #74-412 – the most intensive reflections 330,001 (2q=21.8°) and 390,461 (2q=41.3°) are not present or the cell parameter of the phase is significantly differs from the PDF card. It is hard to perform any conclusions from presented XRD data analysis.
- Concerning the XPS data. The spectra shown in Fig. 6 it is the Auger CuLMM spectra, not the Cu2p core level peak. Why the Zr3d spectra are not presented for the Zr-containing samples. The atomic ratios of detected elements calculated from XPS data should be also presented.
- The conclusions are mainly based on the data of catalytic measurements. It is not really clear which conclusions done using the data obtained during the characterization of the catalysts.
Author Response
please refer to the attachmen.

Reviewer 2 Report
Manuscript ID: molecules-1493022
The research entitled “Dimerization of acetylene to monovinylacetylene (MVA) by bi- 2 metallic Zr/Cu catalyst in Nieuwland catalytic system” dealing about production of MVA from acetone using Zr/Cu. However, the activity and characterization results are not well collaborated and the characterization results are completely wrong. So the manuscript is rejected in the present form. It need to be addressed the following issues before taken in to consideration for publication. The major concerns are
- In experimental section, the labeling of acetyl acetone (C1), acetone (C2), acetic acid (C3), are different from the equation for conversion of acetylene where C1, C2 C3, C4, and C5 represent to other chemical name. Please make a clear distinct with the designation.
- Line no 137; we test the catalytic activities through seven types of catalysts presented in Fig. 1. But in Fig.1 shows only six type of catalysts. Please check this sentence.
- Fig.2. the activity results of C1and C3 are almost similar from 300 min in TOS. The reason for to have best activity of this two reactant is not clear about the active sites whether Zr +4 or Cu+.
- The TPR results of NC and Zr/NC represent two stage of copper reduction (Cu+2 to Cu+ to Cu0 ) from temperature range between 400 to 700 0 Usually bulk copper reduction occur below 200 0C but this results are strange to me. Why did the copper reduction occur above 400 0C both catalysts? Neither Zr role nor reduction of Zr in Zr/NC is explained. Please check one of the article for copper TPR results in Fuel Processing Technology 137 (2015) 220–228.
- XPS results of Cu 2p3/2 are completely wrong explanation. Please check the copper XPS results in one of the article Materials Chemistry and Physics 229 (2019) 402–411.
Reviewer 3 Report
The paper is about the improvement of dimerization of acetylene to monovinylacetylene using a Zr/Cu catalyst.
First, I would say that this paper is of great interest but authors should read again their manuscript and correct all the subscript/superscript missing (ie Cu2+, Zr4+, CuCl24, Cu+ etc...)
Secondly, authors may add a short graph to remind the mechanistic Nieuwland catalytic system to readers, and maybe show their hypotheses on how the adding of Zr implies a better catalytic performance. Indeed, if the catalyst was absolutely finely described, One lacks a proper hypothesis / conclusion on its right mechanism of action.
Round 2
Reviewer 1 Report
The authors attempted to respond to my previous comments without adding new elements to their manuscript. As of such, my opinion of this work has not changed. The first and the fourth questions were basically ignored by the authors.
In responding to my question about the using of the data obtained for dried catalysts in order to make some conclusions concerning the structure and chemical composition of it in dissolved state authors just claimed that there are lack of the techniques for liquids investigation. But this does not confirm the validity of the approach used. Concerning the reduction temperature – the H[CuCl2] from my opinion in presence of water should transform to CuCl and then in principal should oxidized during drying. Anyway it is not clear what is the structure/Cu charging state of Cu112Cl168 phase (the direct using of google search did not give any information)? How CuCl2 could transform to this phase? Again how to prove the presence of this phase in dissolved state? etc. The XPS – still the Zr3d and Atomic ratios between the elements obtained should be included.
Reviewer 2 Report
The manuscript entitled “ Dimerization of acetylene to monovinylacetylene (MVA) by bi- metallic Zr/Cu catalyst in Nieuwland catalytic system” was revised according to the points raised by reviewers. The Authors response is very satisfactory to me . I accept the present form of manuscript to publish in the prestigious general "Molecule" with out further comments.
Round 3
Reviewer 1 Report
The revision submitted by the Authors meets most of the points highlighted by me. I suggests therefore to publish the work as it is, with no further actions from the Authors.